pyVHR: a Python framework for remote photoplethysmography

Boccignone Giuseppe 1
Conte Donatello 2
Cuculo Vittorio 1
D’Amelio Alessandro alessandro.damelio@unimi.it 1
Grossi Giuliano 1
Lanzarotti Raffaella 1
Mortara Edoardo 1
1 PHuSe Lab - Dipartimento di Informatica, Università degli Studi di Milano , Milan , Italy
2 Laboratoire d’Informatique Fondamentale et Appliquée de Tours, Université de Tours , Tours , France
Fernandez-Lozano Carlos
Electronic publication date: 2022 Apr 15
Publication date: 2022
Volume: 8
Electronic Location ID: e929
Received 2021 Oct 26; Accepted 2022 Mar 3
Copyright: ©2022 Boccignone et al.
Copyright year: 2022
Copyright holder: Boccignone et al.
License: This is an open access article distributed under the terms of the Creative Commons Attribution License, which permits unrestricted use, distribution, reproduction and adaptation in any medium and for any purpose provided that it is properly attributed. For attribution, the original author(s), title, publication source (PeerJ Computer Science) and either DOI or URL of the article must be cited.
License URL: https://creativecommons.org/licenses/by/4.0/

Keywords: Remote photoplethysmography, Contactless monitoring, Deepfake Detection, Heart Rate Estimation, Deep rPPG

Funding: The University of Milan through the APC initiative This work was supported by the University of Milan through the APC initiative. The funders had no role in study design, data collection and analysis, decision to publish, or preparation of the manuscript.

==============================
Remote photoplethysmography (rPPG) aspires to automatically estimate heart rate (HR) variability from videos in realistic environments. A number of effective methods relying on data-driven, model-based and statistical approaches have emerged in the past two decades. They exhibit increasing ability to estimate the blood volume pulse (BVP) signal upon which BPMs (Beats per Minute) can be estimated. Furthermore, learning-based rPPG methods have been recently proposed. The present pyVHR framework represents a multi-stage pipeline covering the whole process for extracting and analyzing HR fluctuations. It is designed for both theoretical studies and practical applications in contexts where wearable sensors are inconvenient to use. Namely, pyVHR supports either the development, assessment and statistical analysis of novel rPPG methods, either traditional or learning-based, or simply the sound comparison of well-established methods on multiple datasets. It is built up on accelerated Python libraries for video and signal processing as well as equipped with parallel/accelerated ad-hoc procedures paving the way to online processing on a GPU. The whole accelerated process can be safely run in real-time for 30 fps HD videos with an average speedup of around 5. This paper is shaped in the form of a gentle tutorial presentation of the framework.

Introduction

Heart rate variability can be monitored via photoplethysmography (PPG), an optoelectronic measurement technology first introduced in Hertzman (1937), and then largely adopted due to its reliability and non-invasiveness (Blazek & Schultz-Ehrenburg, 1996). Principally, this technique captures the amount of reflected light skin variations due to the blood volume changes.

Successively, remote-PPG (rPPG) has been introduced. This is a contactless technique able to measure reflected light skin variations by using an RGB-video camera as a virtual sensor (Wieringa, Mastik & Steen, 2005; Humphreys, Ward & Markham, 2007). Essentially, rPPG techniques leverage on the RGB color traces acquired over time and processed to approximate the PPG signal. As a matter of fact, rPPG has sparked great interest by fostering the opportunity for measuring PPG at distance (e.g., remote health assistance) or in all those cases where contact has to be prevented (e.g., surveillance, fitness, health, emotion analysis) (Aarts et al., 2013; McDuff, Gontarek & Picard, 2014; Ramírez et al., 2014; Boccignone et al., 2020b; Rouast et al., 2017). Indeed, the rPPG research field has witnessed a growing number of techniques proposed for making this approach more and more robust and thus viable in contexts facing challenging problems such as subject motion, ambient light changes, low-cost cameras (Lewandowska et al., 2011; Verkruysse, Svaasand & Nelson, 2008; Tarassenko et al., 2014; Benezeth et al., 2018; Wang et al., 2016; Wang, Stuijk & De Haan, 2015; Pilz et al., 2018; De Haan & Van Leest, 2014). More recently, alongside the traditional methods listed above, rPPG approaches based on deep learning (DL) have burst into this research field (Chen & McDuff, 2018; Niu et al., 2019; Yu et al., 2020; Liu et al., 2020; Liu et al., 2021; Gideon & Stent, 2021; Yu et al., 2021).

The blossoming of the field and the variety of the proposed solutions raise the issue, for both researchers and practitioners, of a fair comparison among proposed techniques while engaging in the rapid prototyping and the systematic testing of novel methods. Under such circumstances, several reviews and surveys concerning rPPG (McDuff et al., 2015; Rouast et al., 2018; Heusch, Anjos & Marcel, 2017a; Unakafov, 2018; Wang et al., 2016; McDuff & Blackford, 2019; Cheng et al., 2021; McDuff, 2021; Ni, Azarang & Kehtarnavaz, 2021) have conducted empirical comparisons, albeit suffering under several aspects, as discussed in ‘Related Works’.

To promote the development of new methods and their experimental analysis, in Boccignone et al. (2020a) we proposed pyVHR, a preliminary version of a framework supporting the main steps of the traditional rPPG pulse rate recovery, together with a sound statistical assessment of methods’ performance. Yet, that proposal exhibited some limits, both in terms of code organization, usability, and scalability, and since it was suitable for traditional approaches only. Here we present a new version of pyVHR,1 with a totally re-engineered code, which introduces several novelties.

First of all, we provide a dichotomous view of remote heart rate monitoring, leading to two distinct classes of approaches: traditional methods (section ‘Pipeline for Traditional Methods’) and DL-based methods (section ‘Pipeline for Deep-Learning Methods’). Moreover, concerning the former, a further distinction is setup, concerning the Region Of Interest (ROI) taken into account, thus providing both holistic and patch-based methods. The former takes into account the whole skin region, extracted from the face captured in subsequent frames. Undoubtedly, it is the simplest approach, giving satisfying results when applied on video acquired in controlled contexts. However, in more complex settings the illumination conditions are frequently unstable, giving rise to either high variability of skin tone or shading effects. In these cases the holistic approach is prone to biases altering subsequent analyses. Differently, the patch-based approach employs and tracks an ensemble of patches sampling the whole face. The rationale behind this choice is twofold. On the one hand, the face regions affected by either shadows or bad lighting conditions can be discarded, thus avoiding uncorrelated measurements with the HR ground-truth. On the other hand, the amount of observations available allows for making the final HR estimate more robust, even through simple statistics (e.g., medians), while controlling the confidence levels.

Second, the framework is agile, covers each stage of the pipeline that instantiates it, and it is easily extensible. Indeed, one can freely embed new methods, datasets or tools for the intermediate steps (see section ‘Extending the Framework’) such as for instance: face detection and extraction, pre- and post-filtering of RGBs traces or BVPs signals, spectral analysis techniques, statistical methods.

pyVHR can be easily employed for many diverse applications such as anti-spoofing, aliveness detection, affective computing, biometrics. For instance, in section ‘Case Study: DeepFake detection with pyVHR’ a case study on the adoption of rPPG technology for a Deepfake detection task is presented.

Finally, computations can be achieved in real-time thanks to the NVIDIA GPU (Graphics Processing Units) accelerated code and the use of optimized Python primitives.

Related works

In the last decade the rPPG domain has witnessed a flourish of investigations (McDuff et al., 2015; Rouast et al., 2018; Heusch, Anjos & Marcel, 2017a; Unakafov, 2018; Wang et al., 2016; McDuff & Blackford, 2019; Cheng et al., 2021; McDuff, 2021; Ni, Azarang & Kehtarnavaz, 2021). Yet, the problem of a fair and reproducible evaluation has been in general overlooked. It is undeniable that theoretical evaluations are almost infeasible, given the complex operations or transformations each algorithm performs. Nevertheless, empirical comparisons could be very informative if conducted in the light of some methodological criteria (Boccignone et al., 2020a). In brief: pre/post processing standardization; reproducible evaluation; multiple dataset testing; rigorous statistical assessment.

To the best of our knowledge, a framework respecting all these criteria was missing until the introduction of the early version of pyVHR  Boccignone et al. (2020a).

In Heusch, Anjos & Marcel (2017a) a Python collection of rPPG algorithms is presented, without claiming to be complete in the method assessment.

Interestingly, in Unakafov (2018), the authors highlight the dependency of the pulse rate estimation on five main steps: ROI-selection, pre-processing, rPPG method, post-processing, pulse rate estimation. They present a theoretical framework to assess different pipelines in order to find out which combination provides the most precise PPG estimation; results are reported on the DEAP dataset (Koelstra et al., 2011). Unfortunately, no code has been made available.

In Pilz (2019) a MATLAB toolbox is presented, implementing two newly proposed methods, namely Local Group Invariance (LGI) (Pilz et al., 2018) and Riemannian-PPGI (SPH) (Pilz, 2019), and comparing them to the GREEN channel expectation (Verkruysse, Svaasand & Nelson, 2008) baseline, and two state-of-the-art methods, i.e., Spatial Subspace Rotation (SSR) (Wang, Stuijk & De Haan, 2015), and Projection Orthogonal to Skin (POS) (Wang et al., 2016).

In McDuff & Blackford (2019) the authors propose iPhys, a MATLAB toolbox implementing several methods, such as Green Channel, POS, CHROM (De Haan & Jeanne, 2013), ICA (Poh, McDuff & Picard, 2010), and BCG (Balakrishnan, Durand & Guttag, 2013). The toolbox is presented as a bare collection of method implementations, without aiming at setting up a rigorous comparison framework on one or more datasets. It is worth noticing that all these frameworks are suitable for traditional methods only. Table 1 summarizes at a glance the main differences between pyVHR and the already proposed frameworks.

Table 1 A comparison of the freely available rPPG frameworks.

Check signs mark conditions fulfilled; crosses, those neglected.

	Lang.	Modular	Deep-Ready	Multi-Data	Stat. Assessment	
pyVHR	Python	✓	✓	✓	✓	
McDuff & Blackford (2019)	MATLAB	×	×	×	×	
Heusch, Anjos & Marcel (2017a)	Python	×	×	×	×	
Pilz (2019)	MATLAB	×	×	✓	×	

Installation

The quickest way to get started with pyVHR is to install the miniconda distribution, a lightweight minimal installation of Anaconda Python.

Once installed, create a new conda environment, automatically fetching all the dependencies based on the adopted architecture—with or without GPU—, by one of the following commands:

$  conda  env  create −−f i l e  https://github.com/phuselab/pyVHR/blob/pyVHR_CPU/pyVHR_CPU_env.yml

for CPU-only architecture, or

$  conda  env  create −−f i l e  https://github.com/phuselab/pyVHR/blob/main/pyVHR_env.yml

for a CPU architecture with GPU support. The latest stable release build of pyVHR can be installed, inside the newly created conda environment, with:

$  pip  install  pyvhr−cpu

for CPU-only, or

pip  install  pyvhr

for CPU with GPU support.

The source code for pyVHR can be found on GitHub at https://github.com/phuselab/pyVHR and it is distributed under the GPL-3.0 License. On GitHub, the community can report issues, questions as well as contribute with code to the project. The documentation of the pyVHR framework is available at https://phuselab.github.io/pyVHR/.

pyVHR Pipeline for Traditional Methods

In this section, we introduce the pyVHR modules to be referred by traditional rPPG methods. They are built on top of both APIs developed for the purpose, and open-source libraries. This pipeline follows a software design strategy that assemble sequential modules or stages, with the output of a stage serving as input to one or more subsequent stages. This responds to the need for the framework to be flexible and extensible in order to be more maintainable and improvable over time with innovative or alternative techniques.

The pipeline Stages

The Pipeline() class implements the sequence of stages or steps that are usually required by the vast majority of rPPG methods proposed in the literature, in order to estimate the BPM of a subject, given a video displaying his/her face. Eventually, going through all these steps in pyVHR is as simple as writing a couple of lines of Python code:

from  pyVHR.analysis.pipeline import  Pipeline pipe = Pipeline() time, BPM, uncertainty = pipe.run_on_video('/path/to/vid.avi')

Calling the run_on_video() method of the Pipeline() class starts the analysis of the video provided as argument and produces as output the time step of the estimated BPM and related uncertainty estimate. Figure 1 depicts the predicted BPM on Subject1 of the UBFC2 dataset (Bobbia et al., 2019) (blue trajectory). For comparison, the ground truth BPM trajectory (as recorded from a PPG sensor) is reported in red.

Figure 1 Prediction example.

Predictions on the Subject1 of the UBFC Dataset.

On the one hand the above-mentioned example witnesses the ease of use of the package by hiding the whole pipeline behind a single function call. On the other hand it may be considered too constraining as hinders the user from exploiting its full flexibility. Indeed, the run_on_video() method can be thought of as a black box delivering the desired result with the least amount of effort, relying on default parameter setting.

Nevertheless, some users may be interested in playing along with all the different modules composing the pyVHR pipeline and the related parameters. The following sections aim at describing in detail each of such elements. These are shown in Fig. 2B and can be recapped as follows:

Figure 2 The pyVHR pipeline at a glance.

(A) The multi-stage pipeline of the pyVHR framework for BPM estimate through PSD analysis exploiting end-to-end DL-based methods. (B) The multi-stage pipeline for traditional approaches that goes through: windowing and patch collection, RGB trace computation, pre-filtering, the application of an rPPG algorithm estimating a BVP signal, post-filtering and BPM estimate through PSD analysis.

1. Skin extraction: The goal of this first step is to perform a face skin segmentation in order to extract PPG-related areas; the latter are subsequently collected in either a single patch (holistic approach) or a bunch of “sparse” patches covering the whole face (patch-wise approach).

2. RGB signal processing: The patches, either one or more, are coherently tracked and are used to compute the average colour intensities along overlapping windows, thus providing multiple time-varying RGB signals for each temporal window.

3. Pre-filtering: Optionally, the raw RGB traces are pre-processed via canonical filtering, normalization or de-trending; the outcome signals provide the inputs to any subsequent rPPG method.

4. BVP extraction: The rPPG method(s) at hand is applied to the time-windowed signals, thus producing a collection of heart rate pulse signals (BVP estimates), one for each patch.

5. Post-filtering: The pulse signals are optionally passed through a narrow-band filter in order to remove unwanted out-of-band frequency components.

6. BPM estimation: A BPM estimate is eventually obtained through simple statistics relying on the apical points of the BVP power spectral densities.

Skin extraction

The skin extraction step implemented in pyVHR consists in the segmentation of the face region of the subject. Typically, the regions corresponding to the eyes and mouth are discarded from the analysis. This can be accomplished by pyVHR in two different ways, denoted as:

1. the Convex-hull extractor,

2. the Face parsing extractor.

The Convex-hull extractor considers the skin region as the convex-hull of a set of the 468 facial fiducial points delivered by the MediaPipe face mesh (Lugaresi et al., 2019). The latter provides reliable face/landmark detection and tracking in real-time. From the convex-hulls including the whole face, pyVHR subtracts those computed from the landmarks associated to the eyes and mouth. The resulting mask is employed to isolate the pixels that are generally associated to the skin. An example is shown in the left image of Fig. 3 on a subject of the LG-PPGI dataset (Pilz et al., 2018).3

Figure 3 Comparison of the two implemented skin extraction methods.

Output of the Convex-hull approach (A) and face parsing by BiSeNet (B) on a subject of the LGI-PPGI dataset (Pilz et al., 2018).

Alternatively, the Face parsing extractor computes a semantic segmentation of the subject’s face. It produces pixel-wise label maps for different semantic components (e.g., hair, mouth, eyes, nose, c...), thus allowing to retain only those related to the skin regions. Face semantic segmentation is carried over with BiSeNet (Yu et al., 2018), which supports real-time inference speed. One example is shown in the right image of Fig. 3.

Both extraction methods are handled in pyVHR by the SignalProcessing() class. The following lines of code set-up the extractor with the desired skin extraction procedure:

from  pyVHR.extraction.sig_processing  import  SignalProcessing sig_processing  = SignalProcessing()  i f  skin_method  == 'convexhull':    sig_processing.set_skin_extractor(SkinExtractionConvexHull(target_device))  e l i f  skin_method  == 'faceparsing':    sig_processing.set_skin_extractor(SkinExtractionFaceParsing(target_device))

Holistic approach

The skin extraction method paves the way to the RGB trace computation which is accomplished in a channel-wise fashion by averaging the facial skin colour intensities. This is referred to as the holistic approach, and within the pyVHR framework it can be instantiated as follows:

sig =  sig_processing.extract_holistic(videoFileName)

Patch-based approach

In contrast to the holistic approach, the patch-based one takes into account a bunch of localized regions of interest, thus extracting as many RGB traces as patches. Clearly, imaging photoplethysmography in unconstrained settings is sensitive to subjects changing pose, moving their head or talking. This calls for a mechanism for robust detection and tracking of such regions.

To such end, pyVHR again relies on the MediaPipe Face Mesh, which establishes a metric 3D space to infer the face landmark screen positions by a lightweight method to drive a robust and performant tracking. The analysis runs on CPU and has a minimal speed or memory footprint on top of the inference model.

The user can easily select up to 468 patches centered on a subset of landmarks and define them as the set of informative regions on which the subsequent steps of the pipeline are evaluated. An example of landmark extraction and tracking is shown in Fig. 4. Note that eventually, a patch may disappear due to subject’s movements, hence delivering only partial or none contribution.

Figure 4 Landmarks automatically tracked by MediaPipe and correspondent patch tracking on a subject of the LGI-PPGI dataset (Pilz et al., 2018).

It is worth noting how the user is allowed to arbitrarily compose its own set of patches by exploiting pyVHR utility functions. In the example below, three patches have been selected corresponding to the forehead, left and right cheek areas. Usually, several patches are chosen in order to better control the high variability in the results and to achieve high level of confidence, while making smaller the margin of error.

As for the holistic approach, video loading and patch extraction are handled by few APIs available in the SignalProcessing() class, as shown in the following script.

from  pyVHR.extraction.utils import  MagicLandmarks ldmks_list  = [MagicLandmarks.cheek_left_top[16],  MagicLandmarks.cheek_right_top[14],  MagicLandmarks.forehead_center[1]] sig_processing.set_landmarks(ldmks_list) #  set   squares   patches   side   dimension sig_processing.set_square_patches_side(28.0) #Extract   square   patches  and  compute  the  RGB  t r a j e c t o r i e s   as  the   channel−wise  mean sig =  sig_processing.extract_patches(videoFileName, 'squares', 'mean')

RGB signal computation

In this step, the skin regions detected and tracked on the subject’s face are split in successive overlapping time windows. Next, the RGB traces associated to each region are computed by averaging their colour intensities. More formally, let us consider an RGB video v ∈ ℝw×h×3×T of T frames containing a face, split on P (possibly overlapped) patches. Once the ith patch has been selected, an RGB signal qi(t) is computed. Denote pijtj=1Ni the set of Ni pixels belonging to the ith patch at time t, where pijt∈0,2553. Then, qi(t) is recovered by averaging on pixel colour intensities, i.e., qit=1Ni ∑j=1Nipijt,i=1,…,P.

In the time-splitting process, fixed an integer τ > 0, qi(t) is sliced into K overlapping windows of M = WsFs frames, thus obtaining qikt=qitwt−kτFs,k=0,…,K−1.

where Fs represents the video frame rate, Ws the window length in seconds, while w is the rectangular window defined as: (1) wt=1,0≤t<M0,otherwise.

In order for the signal segments to actually overlap, the overlap inequality τ < Ws must be verified.

Figure 5 shows how the above described patch-based split and tracking procedure is put in place.

Figure 5 Patch tracking within a frame temporal window on a subject of the LGI-PPGI dataset (Pilz et al., 2018).

In pyVHR , the extraction of the windowed RGB signals is computed by the following code snippet.

from  pyVHR.extraction.utils import  sig_windowing,  get_fps Ws = 6 # window  lenght   in   seconds overlap = 1 # window  overlap   in   seconds fps =  get_fps(videoFileName) windowed_sig, timesES =  sig_windowing(sig, Ws, overlap, fps)

Notably, beside being able to switch between convex-hull and face parsing, the user can easily change the main process parameters such as the window length and the amount of frame overlapping.

Methods for BVP estimation

Given that the framework can rely on holistic-wise and patch-wise processing, pyVHR estimates the BVP signal either from a single trace or leveraging on multiple traces. In both cases it employs a wide range of state of the art rPPG methods.

In particular, the windowed RGB traces qikt (i = 1, …, P, with P = 1 in the holistic case) of length K are given in input to the rPPG method at hand, which outputs the signals yikt representing the estimated BVP associated to the ith patch in the k-th time window.

The many methods that have been proposed in the recent literature mostly differ in the way of combining such RGB signals into a pulse-signal. The pool of methods provided by pyVHR , together with a description of the main concepts grounding them, is provided in Table 2. A review of the principles/assumptions behind each of the implemented algorithms is out of the scope of the present work. The interested reader might refer to Wang et al. (2016), McDuff et al. (2015) and Rouast et al. (2018).

Table 2 Traditional rPPG algorithms implemented in pyVHR.

Method	Description	
GREEN (Verkruysse, Svaasand & Nelson, 2008)	The Green (G) temporal trace is directly considered as an estimate of the BVP signal. Usually adopted as a baseline method.	
ICA (Poh, McDuff & Picard, 2010)	Independent Component Analysis (ICA) is employed to extract the pulse signal via Blind Source Separation of temporal RGB mixtures.	
PCA (Lewandowska et al., 2011)	Principal Component Analysis (PCA) of temporal RGB traces is employed to estimate the BVP signal.	
CHROM (De Haan & Jeanne, 2013)	A Chrominance-based method for the BVP signal estimation.	
PBV (De Haan & Van Leest, 2014)	Computes the signature of blood volume pulse changes to distinguish the pulse-induced color changes from motion noise in RGB temporal traces.	
SSR (S2R) (Wang, Stuijk & De Haan, 2015)	Spatial Subspace Rotation (SSR); estimates a spatial subspace of skin-pixels and measures its temporal rotation for extracting pulse signal.	
POS (Wang et al., 2016)	Plane Orthogonal to the Skin (POS). Pulse signal extraction is performed via a projection plane orthogonal to the skin tone.	
LGI (Pilz et al., 2018)	Local Group Invariance (LGI). Computes a feature representation which is invariant to action and motion based on differentiable local transformations.	

Currently, the package implements the following methods for the estimation of the pulse signal from the RGB traces: GREEN (Verkruysse, Svaasand & Nelson, 2008), CHROM (De Haan & Jeanne, 2013), ICA (Poh, McDuff & Picard, 2010), LGI (Pilz et al., 2018), PBV (De Haan & Van Leest, 2014), PCA (Lewandowska et al., 2011), POS (Wang et al., 2016), SSR (Wang, Stuijk & De Haan, 2015). However, the user may define any custom method for estimating BVP by extending the pyVHR.BVP.methods module.

The BVP signal can be estimated in pyVHR as follows:

     bvp =  RGB_sig_to_BVP(windowed_sig, fps, method=cpu_POS)

Figure 6 depicts the BVP signals estimated by four different rPPG methods implemented in pyVHR (POS, GREEN, CHROM, PCA), on the same time window using the holistic patch.

Figure 6 Predicted BVP signals.

An example of estimated BVP signals on the same time window by four different methods. (A) POS. (B) GREEN. (C) CHROM. (D) PCA.

Pre and post-filtering

pyVHR offers simple APIs to apply filters on either the RGB traces qi(t) (pre-filtering) or the estimated pulse signal yi(t) (post-filtering). A set of ready to use filters are implemented, namely:

• Band Pass (BP) filter: filters the input signal using a bandpass N-th order Butterworth filter with a given passband frequency range.

• Detrending: subtracts offsets or linear trends from time-domain input data.

• Zero-Mean: Removes the DC component from a given signal.

However, the user can adopt any custom filter complying with the function signature defined in pyVHR.BVP.filters. The following provides an example of how to detrend an RGB trace qi(t):

filtered_sig  =  apply_filter(sig, detrend)

Additionally, a Band-Pass filter can be applied on the estimated BVP signals yi(t) in order to rule out the frequencies that leave outside the feasible range of typical heart rates (which is usually between 40 Hz and 200 Hz):

filtered_bvp  =  apply_filter(bvp, BPfilter,                  params={'order':6,'minHz':0.65,'maxHz':4.0,'fps':fps})

From BVP to BPM

Given the estimated BVP signal, the beats per minute (BPM) associated to a given time window can be easily recovered via analysis of its frequency domain representation. In particular, pyVHR estimates the Power Spectral Density (PSD) of the windowed pulse signal yikt via discrete time Fourier transform (DFT) using the Welch’s method. The latter employs both averaging and smoothing to analyze the underlying random process.

Given a sequence yikt, call Sikν its power spectra (periodogram) estimated via the Welch’s method. The BPM is recovered by selecting the normalized frequency associated to the peak of the periodogram: ν ˆik= argmaxν∈ΩSikν,

corresponding to the PSD maxima as computed by Welch’s method on the range Ω = [39, 240] of feasible BPMs.

The instantaneous BPM associated to the k-th time window (k ∈ 1, …, K) for the ith patch (i ∈ 1, …, P), is recovered by converting the normalized peak frequency ν ˆik into an actual frequency, h ˆik=ν ˆikFsL,

where Fs is the video frame rate and L is the DFT size. Figure 7 shows the Welch’s estimates for the BVP signals of Fig. 6. The peak in the spectrum represents the instantaneous Heart Rate (h ˆik).

Figure 7 Estimated PSD.

Estimated Power Spectral Densities (PSD) for the BVP signals plotted in Fig. 6. The BPM estimate, given by the maxima of the PSD, is represented by the blue dashed line. (A) POS. (B) GREEN. (C) CHROM. (D) PCA.

When multiple patches have been selected (P > 1), the predicted BPM for the k-th time window can be obtained resorting to simple statistical measures. Specifically, pyVHR computes the median BPM value of the predictions coming from the P patches.

Formally, call Hk the ordered list of P BPM predictions coming from each patch in the k-th time window; then: (2) h ˆk=medianHk=HkP−12ifPis oddHkP2−1+HkP22ifPis even.

Note that if the number of patches P = 1 (i.e., a single patch has been selected or the holistic approach has been chosen), then: (3) h ˆk=Hk0.

Moreover, when multiple patches have been selected, a measure of variability of the predictions can be computed in order to quantify the uncertainty of the estimation. In particular, pyVHR computes the Median Absolute Deviation (MAD) as a robust measure of statistical dispersion. The MAD is defined as: (4) MADk=median|Hk−h ˆk|.

Clearly, the MAD drops to 0 when P = 1. Figure 8 depicts the distribution of predicted BPM in a given time window, when P = 100 patches are employed. The results from different methods are shown for comparison. Note how the median is able to deliver precise predictions, while the MAD represents a robust measure of uncertainty.

Figure 8 Distribution of BPM predictions by four methods on P patches.

(A) POS. (B) GREEN. (C) CHROM. (D) PCA. Kernel Density Estimates (KDEs) of the predicted BPMs in a time window from P = 100 patches. The ultimate BPM prediction is given by the median (gold dashed line). The uncertainty estimate delivered by the Median Absolute Deviation (MAD) is shown by the golden band around the median. The blue dashed line represents the actual BPM.

Computing the BPM from the BVP signal(s) can be easily accomplished in pyVHR as follows:

from  pyVHR.BPM.BPM import  BVP_to_BPM ,  multi_est_BPM_median bpmES =  BVP_to_BPM(bvp, fps) #  median  BPM  from   multiple   estimators  BPM bpm, uncertainty =  multi_est_BPM_median(bpmES)

The result along with the ground-truth are shown in Fig. 9.

Figure 9 Comparison of predicted vs ground truth BPMs using the patch-wise approach.

Predicted BPM (blue) for the Subject1 of the UBFC Dataset. The uncertainty is plotted in shaded blue, while the ground truth is represented by the red line.

Efficient computation and GPU acceleration

Most of the steps composing the pipeline described above are well suited for parallel computation. For instance, the linear algebra operations involved in the pulse signal recovery from the RGB signal or, more generally, the signal processing steps (e.g., filtering, spectral estimation, etc.), not to mention the skin segmentation procedures from high resolution videos.

To such end, pyVHR exploits the massive parallelism of Graphical Processing Units (GPUs). It is worth mentioning that GPUs are not strictly required to run pyVHR code; nevertheless, in some cases, GPU accelerated code allows to run the pipeline in real-time.

Figure 10 shows the average per-frame time requirement for getting through the whole pipeline when using the POS method. It is worth noticing that, when using the Holistic approach (or equivalently one single patch), a video frame can be processed in less than 0.025 seconds, regardless of the adopted architecture (either CPU or GPU). This means that the whole pipeline can be safely run in real-time for videos at 30 frames per second (the 30 fps time limit is represented by the dashed green line).

Figure 10 Per-frame time requirements.

Average time requirements to process one frame by the Holistic and Patches approaches when using CPU vs. GPU accelerated implementations. The green dashed line represents the real-time limit at 30 frames per second (fps).

Obviously, when multiple patches are employed (in the example of Fig. 10, P = 100 patches are used), the average time required by CPUs to process a single frame rises up to about 0.12 seconds. Notably, the adoption of GPU accelerated code allows to run the whole pipeline in real-time, even when using a huge number of patches. Indeed, the ratio to CPU time and GPU time, i.e., the speedup defined as timeseq/timeparall, is about 5. Remarkably, similar gain in performances are observed if adopting any other rPPG method.

The result shown in Fig. 10 refers to the following hardware configuration: Intel Xeon Silver 4214R 2.40 GHz (CPU), NVIDIA Tesla V100S PCIe 32GB (GPU). Similar results were obtained relying on a non-server configuration: Intel Core i7-8700K 4.70 GHz (CPU), NVIDIA GeForce GTX 960 2GB (GPU). The maximum RAM usage for 1 min HD video analysis is 2.5 GB (average is 2 GB); the maximum GPU memory usage for 1 min HD video analysis is 1.8 GB (average is 1.4 GB).

In the following it is shown how to enable CUDA GPU acceleration on different steps in the Pipeline:

• Skin extraction: Convex Hull and Face Parsing. The user can easily choose to run this step with CPU or GPU:      target_device  = 'GPU' #  or   ’CPU’        sig_processing  = SignalProcessing()      sig_processing.set_skin_extractor(                  SkinExtractionConvexHull(target_device))      sig_processing.set_skin_extractor(                  SkinExtractionFaceParsing(target_device))

• rPPG Methods: the package contains different version of the same method. For example the CHROM method is implemented for both CPU and GPU.      bvp =  RGB_sig_to_BVP(sig, fps,  device_type='cuda', method=cupy_CHROM)      bvp =  RGB_sig_to_BVP(sig, fps,  device_type='cpu', method=cpu_LGI)

• BPM Estimation:        bpmES =  BVP_to_BPM_cuda(bvp, fps)    # GPU            bpmES =  BVP_to_BPM(bvp, fps)          # CPU

GUI for online processing

Besides being used as a Python library, pyVHR makes available a Graphical User Interface (GUI). It provides access to most of the available functionalities, while showing the BPMs estimation process in real-time. It is straightforward to use and it allows for setting up the pipeline parameters and the operating mode, by choosing either a webcam or a video file.

To start the GUI, one can run the command:

     $  Python  pyVHR/realtime/GUI.py

Figure 11 shows a screenshot of the GUI during the online analysis of a video. On the top right are presented the video file name, the video FPS, resolution, and a radio button list to select the type of frame displayed. The original or segmented face can be visualized either selecting the Original Video or the Skin option, while the Patches radio button enables the visualization of the patches (in red). The Stop button ends the analysis, and results can be saved on disk by pushing the Save BPMs button.

Figure 11 The graphical user interface.

A screenshot of the graphical user interface (GUI) for online video analysis. The plot on the left shows the predicted BPMs, while on the right it is shown the processed video frames (captured with a webcam) with an example of the segmented skin and the tracked patches.

pyVHR Pipeline for Deep-Learning Methods

Recent literature in computer vision has given wide prominence to end-to-end deep neural models and their ability to outperform traditional methods requiring hand-crafted feature design. In this context, learning frameworks for recovering physiological signals were also born (Chen & McDuff, 2018; Niu et al., 2019; Yu et al., 2020; Yu et al., 2021; Gideon & Stent, 2021; Liu et al., 2021; Nowara, McDuff & Veeraraghavan, 2020). The end-to-end nature of the DL based approaches is reflected by a much simpler pipeline; indeed, these methods typically require as input raw video frames that are processed by the DL architecture at hand and produce either a BVP signal or the estimated heart rate, directly. Figure 2A depicts at a glance the flow of stages involved in the estimation of heart rate using DL based approaches. Clearly, this gain in simplicity comes at the cost of having to train the model on huge amounts of data, not to mention the issues related to the assessment of the model’s generalization abilities.

In the last few years the literature has witnessed a flourish of DL-based approaches(for two recent reviews see Cheng et al. (2021) and Ni, Azarang & Kehtarnavaz (2021). Nonetheless, despite the claimed effectiveness and superior performances, few solutions have been made publicly available (both in terms of code and learned model weights). This raises issues related to proper reproducibility of the results and the method assessment. For instance, a recent efficient neural architecture called MTTS-CAN has been proposed in Liu et al. (2020) being a valuable contribution since the pre-trained model and code are released. It essentially leverages a tensor-shift module and 2D-convolutional operations to perform efficient spatial temporal modeling in order to enable real-time cardiovascular and respiratory measurements. MTTS-CAN can be framed as an end-to-end model since it does not need any pre-processing step before data is fed into the network, except performing trivial image normalizations. MTTS-CAN is included in the pyVHR framework, and below it is shown how practical is to extend the framework with similar DL-based approaches provided that the pre-trained model is available.

As for the pipeline for traditional methods shown in previous section, pyVHR also defines a sequence of stages that allows to recover the time varying heart rate from a sequence of images displaying a face. These are detailed in the following.

The stages for end-to-end methods

Given a video displaying a subject face, the DeepPipeline() class performs the necessary steps for the rPPG estimate using a chosen end-to-end DL method. Specifically, the pipeline includes the handling of input videos, the estimation from the sequence of raw frames and, eventually, the pre/post-processing steps. The following code snippet carries out the above procedure with few statements:

from  pyVHR.analysis.pipeline import  DeepPipeline pipe = DeepPipeline() time, BPM = pipe.run_on_video('/path/to/vid.avi', method='MTTS_CAN')

Figure 2A summarizes the steps involved in a run_on_video() call on a given input video. As in the pipeline using traditional methods (see section ‘Pipeline for Traditional Methods’), after a predetermined chain of analysis steps it produces as output the estimated BPM and related timestamps (time).

For instance, consider the MTTS-CAN model currently embedded into the DeepPipeline() class; it estimates the rPPG pulse signal from which the BPM computation can be carried out by following the very same procedure outlined in section ‘From BVP to BPM’, namely time windowing and spectral estimation. Eventually, some optional pre/post filtering operations (section ‘Pre and Post-Filtering’) can be performed.

The following few lines of Python code allow to carry out the above steps explicitly:

from  pyVHR.extraction.sig_processing  import  SignalProcessing from  pyVHR.extraction.utils import  get_fps from  pyVHR.BPM import  BVP_to_BPM from  pyVHR.utils.errors import  BVP_windowing sp = SignalProcessing() frames = sp.extract_raw('/path/to/videoFileName') fps =  get_fps('/path/to/videoFileName') bvps_pred  =  MTTS_CAN_deep(frames, fps) winsize = 6 # 6∖ s  long   time  window bvp_win, timesES =  BVP_windowing(bvp_pred, winsize, fps, stride=1) bpm =  BVP_to_BPM(bvp_win, fps)

In order to embed a new DL-method, the code above should be simply modified substituting the function MTTS_CAN_deep with a new one implementing the method at hand, while respecting the same signature (cfr. ‘Extending the Framework’).

Assessment of rPPG Methods

Does a given rPPG algorithm outperforms the existing ones? To what extent? Is the difference in performance significantly large? Does a particular post-filtering algorithm cause an increase/drop of performance?

Answering all such questions, calls for a rigorous statistical assessment of rPPG methods. As a matter of fact, although the field has recently experienced a substantial gain in interest from the scientific community, it is still missing a sound and reproducible assessment methodology allowing to gain meaningful insights and delivering best practices.

By and large, novel algorithms proposed in the literature are benchmarked on non-publicly available datasets, thus hindering proper reproducibility of results. Moreover, in many cases, the reported results are obtained with different pipelines; this makes it difficult to precisely identify the actual effect of the proposed method on the final performance measurement.

Besides that, the performance assessment mostly relies on basic and common-sense techniques, such as roughly rank new methods with respect to the state-of-the-art. These crude methodologies often make the assessment unfair and statistically unsound. Conversely, a good research practice should not limit to barely report performance numbers, but rather aiming at principled and carefully designed analyses. This is in accordance with the growing quest for statistical procedures in performance assessment in many different fields, including machine learning and computer vision (Demšar, 2006; Benavoli et al., 2017; Torralba & Efros, 2011; Graczyk et al., 2010; Eisinga et al., 2017).

In the vein of its forerunner (Boccignone et al., 2020a), pyVHR deals with all such problems by means of its statistical assessment module. The design principles can be recapped as follows:

• Standardized pipeline: When setting up an experiment to evaluate a new rPPG algorithm, the whole pipeline (except the algorithm) should be held fixed.

• Reproducible evaluation: The evaluation protocol should be reproducible. This entails adopting publicly available datasets and code.

• Comparison over multiple datasets: In order to avoid dataset bias, the analysis should be conducted on as many diverse datasets as possible.

• Rigorous statistical assessment: The reported results should be the outcome of proper statistical procedures, assessing their statistical significance.

The workflow of the Statistical Assessment Module is depicted in Fig. 12.

Figure 12 The assessment module at a glance.

One or more datasets are loaded; videos are processed by the pyVHR pipeline while ground-truth BPM signals are retrieved. Predicted and real BPM are compared with standard metrics and the results are rigorously analyzed via hypothesis testing procedures.

In a nutshell, each video composing a particular rPPG dataset is processed by the pyVHR pipeline as described above. Moreover, the package provides primitives for loading and processing real BVP signals as recorded from pulse-oximeters. Such signals undergo a treatment similar to the estimated BVP. In particular, the original BVP signal g(t) is sliced into overlapping time windows; for each window the ground truth BPM hk (the BPM associated to the k-th time window, with k = 1, …, K) is recovered via maximization of the Power Spectral Density (PSD) estimate provided by the Welch’s method.

Finally, the estimated (h ˆk) and ground truth (hk) BPM signals are compared with one another exploiting standard metrics (c.f.r ‘Metrics’). Eventually, statistically rigorous comparisons can be effortlessly performed (c.f.r ‘Significance Testing’).

Notably, the many parameters that make up each step of the pipeline (from the ROI selection method to the pre/post filtering operations, passing through the BVP estimation by one or multiple rPPG algorithms) can be easily specified in a configuration (.cfg) file. Setting up a .cfg file allows to design the experimental procedure in accordance with the principles summarized above. A brief description of the implemented comparison metrics and the .cfg file specifications are provided in the following Sections.

Metrics

pyVHR provides common metrics to evaluate the performance of one or more rPPG methods in estimating the correct heart rate (BPM) over time. These are briefly recalled here.

In order to measure the accuracy of the BPM estimate h ˆ, this is compared to the reference BPM as recovered from contact BVP sensors h. To this end, the reference BVP signal g(t) is splitted into overlapping windows, similarly to the procedure described in section ‘Methods for BVP estimation’ for the estimated BVP, thus producing K windowed signals gk (k ∈ 1, …, K). The reference BPM is found via spectral analysis of each window, as described in section ‘From BVP to BPM’. This yields the K reference BPM hk to be compared to the estimated one h ˆk by adopting any of the following metrics:

Mean Absolute Error (MAE). The Mean Absolute Error measures the average absolute difference between the estimated h ˆ and reference BPM h. It is computed as: MAE=1K∑k|h ˆk−hk|.

Root Mean Squared Error (RMSE). The Root-Mean-Square Error measures the difference between quantities in terms of the square root of the average of squared differences, i.e., RMSE=1K∑kh ˆk−hk2.

Pearson Correlation Coefficient (PCC). Pearson Correlation Coefficient measures the linear correlation between the estimate h ˆ and the ground truth h. It is defined as: PCC=∑kh ˆk−μ ˆhk−μσ1σ2,

here μ ˆ and µdenote the means of the respective signals, while σ1 and σ2 are their standard deviations.

Concordance Correlation Coefficient (CCC). The Concordance Correlation Coefficient (Lawrence & Lin, 1989) is a measure of the agreement between two quantities. Like Pearson’s correlation, CCC ranges from -1 to 1, with perfect agreement at 1. It is defined as: CCC=2σ12μ ˆ−μ2+σ12+σ22

where μ ˆ and µdenote the means of the prediceted and reference BPM traces, respectively. Likewise, σ1 and σ2 are their standard deviations, while σ12 is their covariance.

Signal to Noise Ratio (SNR). The SNR (De Haan & Jeanne, 2013) measures the ratio of the power around the reference HR frequency plus the first harmonic of the estimated pulse-signal and the remaining power contained in the spectrum of the estimated BVP. Formally it is defined as: (5) SNR=1K∑K10log10 ∑vUkvSkv2 ∑v1−UkvSkv2

where Sk(v) is the power spectral density of the estimated BVP in the k-th time window and Uk(v) is a binary mask that selects the power contained within ±12 BPM around the reference Heart Rate and its first harmonic.

The configuration (.cfg) file

The .cfg file allows to set up the experimental procedure for the evaluation of models. It is structured into 6 main blocks that are briefly described here:

Dataset. This block contains the information relative to a particular rPPG dataset, namely its name, and its path.           [DATASET]           dataset      = DatasetName           path = None           videodataDIR= /path/to/vids/           BVPdataDIR   = /path/to/gt/           ...

Filters. It defines the filtering methods to be eventually used in the pre/post filtering phase. In the following example a band-pass butterworth filter of 6-th order is defined, with a passing band between 40 Hz and 240 Hz.           [BPFILTER]           path = None           name = BPfilter           params = {'minHz':0.65, 'maxHz':4.0, 'fps':'adaptive', 'order':6}

RGB Signal. Defines all the parameters for the extraction of the RGB signal (e.g., ROI selection method, temporal windowing size, number and type of patches to be used, etc.).        [SIG]           ...           winSize = 6           skin_extractor  = convexhull           approach = patches           ...

BVP. Sets up the rPPG method to be adopted for the estimation of the BVP signal. Multiple methods can be provided in order to compare them. In this example two methods will be analyzed, namely POS and GREEN (adopting their CPU implementations).           [BVP]           methods = ['POS', 'GREEN']           ...

Methods. It allows to configure each rPPG method to be analyzed (e.g., eventual parametrs and pre/post filters). The two methods chosen above are configured here. In particular, POS will not employ any pre/post filtering, while for the GREEN method, the above-defined band pass filter will be applied for both pre and post filtering.           [POS]           ...           name =  cpu_POS           device_type  = cpu           pre_filtering  = []           post_filtering  = []           [GREEN]           ...           name =  cpu_GREEN           device_type  = cpu           pre_filtering  = ['BPFILTER']           post_filtering  = ['BPFILTER']

The experiment on the dataset defined in the .cfg file can be simply launched as:

from  pyVHR.analysis.pipeline import  Pipeline pipe = Pipeline() results = pipe.run_on_dataset('/path/to/config.cfg') results.saveResults("/path/to/results.h5")

In the above code, the run_on_dataset method from the Pipeline class, parses the .cfg file and initiates a pipeline for each rPPG method defined in it. The pipelines are used to process each video in the dataset. Concurrently, ground truth BPM data is loaded and comparison metrics are computed w.r.t. the predictions (cfr. Figure 12). The results are delivered as a table containing for each method the value of the comparison metrics computed between ground truth and predicted BPM signals, on each video belonging to the dataset, which are then saved to disk. The same considerations hold for the definition of .cfg files associated to DL-based methods. Clearly, in this case the information related to the RGB Signal block are unnecessary.

Significance testing

Once the comparison metrics have been computed for all the considered methods, the significance of the differences between their performance can be evaluated. In other words, we want to ensure that such difference is not drawn by chance, but it represents an actual improvement of one method over another.

To this end, pyVHR resorts to standard statistical hypothesis testing procedures. Clearly, the results eventually obtained represent a typical repeated measure design, in which two or more pipelines are compared on paired samples (videos). A number of statistical tests are available in order to deal with such state of affairs.

In the two populations case, typically, the paired t-test is employed; alternatively some non-parametric versions of the paired t-test are at hand, namely the Sign Test or the Wilcoxon signed ranks Test; in general the latter is preferred over the former due to its higher power. For the same reason it is recommended to adopt the parametric paired t-test instead of the non-parametric Wilcoxon test. However, the use of the paired t-test is subject to the constraint of normality of the populations. If such condition is not met, a non-parametric test should be chosen.

Similarly, with more than two pipelines, repeated measure ANOVA is the parametric test that is usually adopted. Resorting to ANOVA, requires Normality and Heteroskedasticity (equality of variances) conditions to be met. Alternatively, when these cannot be ensured, the Friedman Test is chosen.

In pyVHR the Normality and Heteroskedasticity conditions are automatically checked via the Shapiro–Wilk Normality test and, depending on the Normality with Levene’s test or Bartlett’s tests for homogeneity of the data.

In the case of multiple comparisons (ANOVA/Friedman), a proper post-hoc analysis is required in order to establish the pairwise differences among the pipelines. Specifically, the Tukey post-hoc Test is adopted downstream to the rejection of the null hypothesis of ANOVA (the means of the populations are equal), while the Nemenyi post-hoc Test is used after the rejection of the Friedman’s null hypothesis of equality of the medians of the samples.

Besides the significance of the differences, it is convenient to report their magnitude, too. The effect size can be computed via the Cohen’s d in case of Normal of populations; the Akinshin’s γ is used otherwise.

The two populations case

pyVHR automatically handles the above significance testing procedure within the StatAnalysis() class, by relying on the Autorank Python package (Herbold, 2020). StatAnalysis() ingests the results produced at the previous step and runs the appropriate statistical test on a chosen comparison metric:

from  pyVHR.analysis.stats import  StatAnalysis st = StatAnalysis("/path/to/results.h5") # −− box   p l o t   s t a t i s t i c s   ( medians ) st.displayBoxPlot(metric='CCC') #t e s t i n g st.run_stats(metric='CCC')

The output of the statistical testing procedure is reported as follows:

The Shapiro–Wilk Test rejected the null hypothesis of normality for the populations POS (p < 0.01) and GREEN (p < 0.01). (…) the Wilcoxon’s signed rank test has been chosen to determine the differences in the central tendency; median (MD) and median absolute deviation (MAD) are reported for each population. The test rejected the null hypothesis (p < 0.01) that population POS (MD = 1.344 ± 1.256, MAD = 0.688) is not greater than population GREEN (MD = 2.297 ± 3.217, MAD = 1.429). Hence, we assume that the median of POS is significantly larger than the median of GREEN with a large effect size (γ =  − 0.850).

As it can be observed, the appropriate statistical test for two non-normal populations has been properly selected. The Concordance Correlation Coefficient (CCC) for the method POS turned out to be significantly larger than the CCC of the method GREEN. Besides being significant, such difference is substantial, as witnessed by the large effect size.

The more-than-two populations case

Suppose now to structure the above .cfg in order to run three methods instead of two. This would be as simple as extending the “BVP” and “Methods” blocks as follows:

     # ## BVP ###        [BVP]      methods = ['POS', 'GREEN', 'CHROM']      ...      # ## METHODS ###        [POS]      ...      [GREEN]      ...      [CHROM]      ...      name = CHROM      pre_filtering  = ['BPFILTER']      post_filtering  = ['BPFILTER']

Re-running the statistical analysis would yield the following output:

The Shapiro–Wilk Test rejected the null hypothesis of normality for the populations CHROM (p < 0.01), POS (p < 0.01), and GREEN (p < 0.01). Given that more than two populations are present, and normality hypothesis has been rejected, the non-parametric Friedman test is chosen to inspect the eventual significant differences between the medians of the populations. The post-hoc Nemenyi test is then used to determine which differences are significant. The Friedman test rejected the null hypothesis (p < 0.01) of equality of the medians of the populations CHROM (MD = 1.263 ± 1.688, MAD = 0.515, MR = 1.385), POS (MD = 1.344 ± 1.513, MAD = 0.688, MR = 1.769), and GREEN (MD = 2.297 ± 4.569, MAD = 1.429, MR = 2.846). (…) the post-hoc Nemenyi test revealed no significant differences within the following groups: CHROM and POS, while other differences are significant.

Notably, the presence of more than two non-normal populations leads to the choice of the non-parametric Friedman Test as omnibus test to determine if there are any significant differences between the median values of the populations.

The box-plots showing the distributions of CCC values for all methods on the UBFC dataset is provided in Fig. 13, while the output of the post-hoc Nemenyi test can be visualized through the Critical Difference (CD) diagram (Demšar, 2006) shown in Fig. 14; CD Diagrams show the average rank of each method (higher ranks meaning higher average scores); models whose difference in ranks does not exceed the CDα (α = 0.05) are joined by thick lines and cannot be considered significantly different.

Figure 13 Box plots showing the CCC values distribution for the POS, CHROM and GREEN methods on the UBFC2 dataset.

Figure 14 Results of the statistical assessment procedure.

CD diagram displaying the results of the Nemenyi post-hoc test on the three populations (POS, CHROM and GREEN) of CCC values on the UBFC2 dataset.

Comparing deep and traditional pipelines

How does a given DL-based rPPG method compares to the above mentioned traditional approaches? The following code snippet allows to run both the traditional and deep pipelines. The results are saved to the same folder, which is then fed as input to the StatAnalysis class; the join_data =True flag allows to merge the results yielded by the two pipelines, thus enabling the statistical comparison between the chosen methods.

from  pyVHR.analysis.stats import  StatAnalysis from  pyVHR.analysis.pipeline import  Pipeline, DeepPipeline #Pipeline   for   Traditional  Methods traditional_pipe  = Pipeline() traditional_results  =  traditional_pipe.run_on_dataset('/path/to/trad_config.cfg') traditional_results.saveResults("/path/to/results_folder/traditional_results.h5") #Pipeline   for  Deep  Methods deep_pipe  = DeepPipeline() deep_results  =  deep_pipe.run_on_dataset('/path/to/deep_config.cfg') deep_results.saveResults("/path/to/results_folder/deep_results.h5") #S t a t i s t i c a l   Analysis st = StatAnalysis("/path/to/results_folder/", join_data=True) # −− box   p l o t   s t a t i s t i c s st.displayBoxPlot(metric='SNR') #Significance   t e s t i n g  on  the  SNR  metric st.run_stats(metric='SNR')

In this case, the Signal-to-Noise Ratio (SNR) has been chosen as comparison metric; Fig. 15 qualitatively displays the results of the comparison of the above mentioned traditional methods with the MTTS-CAN DL-based approach (Liu et al., 2020). The outcome of the statistical assessment is shown in the CD diagram of Fig. 16.

Figure 15 Box plots showing the SNR values distribution for the POS, CHROM, MTTS-CAN and GREEN methods on the UBFC1 dataset.

Figure 16 Results of the statistical assessment procedure.

CD diagram displaying the results of the Nemenyi post-hoc test on the four populations (POS, CHROM, MTTS-CAN and GREEN) of SNR values on the UBFC1 dataset.

Extending the Framework

Besides assessing built-in methods on public datasets included in the framework, the platform is conceived to allow the addition of new methods or datasets. This way, it is possible to assess a new proposal, comparing it against built-in methods, and testing it on either already included datasets or on new ones, this exploiting all the pre- and post-processing modules made available in pyVHR . The framework extension can be achieved following simple steps as described in the subsequent subsections.

Adding a new method

In this section we show how to add to the pyVHR framework either a new traditional or learning-based method called MY_NEW_METHOD.

In the first case, to exploit the pyVHR built-in modules the new function should receive as input a signal in the shape produced by the built-in pre-processing modules, together with some other parameters required by the method itself. Specifically, this results in a signature of the form:

MY_NEW_METHOD(signal, ∗∗kargs)

where signal is a Numpy array in the form (P, 3, K); P is the number of considered patches (it can be 1 if the holistic approach is used), 3 is the number of RGB Channels and K is the number of frames. **kargs refers to a dictionary that contains all the parameters required by the method at hand. A proper function implementing an rPPG method must return a BVP signal as a Numpy array of shape (P,K).

In case of DL-based method, the new function should receive as input the raw frames as a Numpy array in the form (H,W,3,K), where H,W denote the frame dimensions. The output of the new method could be either a BVP signal or the HR directly.

Accordingly, the signature becomes:

MY_NEW_METHOD(frames, fps)

Both for traditional and DL-based method, the function call MY_NEW_METHOD can now be embedded into the proper Pipeline, and assessed as described earlier. In order to do so, the .cfg file should be tweaked as follows:

[MY_NEW_METHOD] path = 'path/to/module.py' name = 'MY_NEW_METHOD' ...

Moreover, the methods block of the .cfg file is supposed to contain a specific listing describing MY_NEW_METHOD, providing the path to the Python module encoding the method and its function name.

Adding a new dataset

Currently pyVHR provides APIs for handling five datasets commonly adopted for the evaluation of rPPG methods, namely LGI-PPGI (Pilz et al., 2018), UBFC (Bobbia et al., 2019), PURE (Stricker, Müller & Gross, 2014), MAHNOB-HCI (Soleymani et al., 2011), and COHFACE (Heusch, Anjos & Marcel, 2017a). However, the platform allows to add new datasets favoring the method assessment on new data. A comprehensive list of the datasets that are typically employed for rPPG estimation and evaluation is reported in Table 3.

Table 3 A list of datasets commonly used for rPPG.

The left-most column collects the dataset names and introducing papers; second column, the number of subjects involved; third column, the task or condition under which data have been collected (Stationary: subject are asked to sit still; Interaction: emulation of a human–computer interaction scenario via a time sensitive mathematical game; Multiple: more than one condition has been considered while recording subjects, such as Steady, Talking, Head Motion etc; Physical Activities: subjects are recorded while performing activities such as speaking, rowing, exercising on a stationary bike etc; Stress Test: participants are subject to tasks with different levels of difficulty inspired by the Trier Social Stress Test; Emotion Elicitation: participants were shown fragments of movies and pictures apt at eliciting emotional reactions). In the last column, datasets whose handling APIs are currently available in pyVHR have been checked.

Dataset	Subjects	Task/Condition	pyVHR	
UBFC1 (Bobbia et al., 2019)	8	Stationary	✓	
UBFC2 (Bobbia et al., 2019)	42	Interaction	✓	
PURE (Stricker, Müller & Gross, 2014)	10	Multiple	✓	
LGI-PPGI (Pilz et al., 2018)	25 (6 available)	Multiple	✓	
MAHNOB-HCI (Soleymani et al., 2011)	27	Emotion elicitation	✓	
COHFACE (Heusch, Anjos & Marcel, 2017b)	40	Stationary	✓	
UBFC-Phys (Sabour et al., 2021)	56	Stress test	×	
AFRL (Estepp, Blackford & Meier, 2014)	25	Multiple	×	
MMSE-HR (Zhang et al., 2016)	140	Simulating facial expressions	×	
OBF (Li et al., 2018)	106	Multiple	×	
VIPL-HR (Niu et al., 2018)	107	Multiple	×	
ECG-Fitness (Špetlík, Franc & Matas, 2018)	17	Physical activities	×	

The framework conceives datasets as a hierarchy of classes (see Fig. 17) that allows to describe a new dataset by inheriting from the Dataset base class and implementing few methods for loading videos and ground truth PPG data.

Specifically, the following two functions should be supplied:

• a loadFilenames() function to load video files in a Python list; this function has no inputs and defines two class variables, namely videoFilenames and BVPFilenames. These are both Python lists containing, respectively, video and ground-truth BVP filenames from the dataset);

• a readSigfile(filename) function loading and returning the ground-truth BVP signal given a video filename.

The new dataset can then be included in the testing via the .cfg file as described in the paragraph Dataset of section ‘The configuration (.cfg) file’. As for the addition of new method, also in case of adding a new dataset the .cfg file should be completed by specifying the path pointing to the new dataset class:

          [DATASET]           dataset      = DatasetName           path = /path/to/datasetClass/           videodataDIR= /path/to/videos/           BVPdataDIR   = /path/to/gtfiles/           ...

Case Study: DeepFake detection with pyVHR

DeepFakes are a set of DL based techniques allowing to create fake videos by swapping the face of a person by that of another. This technology has many diverse applications such as expression re-enactment (Bansal et al., 2018) or video de-identification (Bursic et al., 2021). However, in recent years the quality of deepfakes has reached tremendous levels of realism, thus posing a series of treats related to the possibility of arbitrary manipulation of identity, such as political propaganda, blackmailing, and fake news (Mirsky & Lee, 2021).

As a consequence, efforts have been devoted to the study and the development of methods allowing to discriminate between real and forged videos (Tolosana et al., 2020; Mirsky & Lee, 2021). Interestingly enough, one effective approach is represented by the exploitation of physiological information (Hernandez-Ortega et al., 2020; Ciftci, Demir & Yin, 2020; Qi et al., 2020) . Indeed, signals originating from biological action such as heart beat, blood flow, or breathing are expected to be (in large part) disrupted after face-swapping. Therefore, methods such as remote PPG can be adopted in order to evaluate their presence.

In the following, it is shown how pyVHR can be effectively employed to easily perform a DeepFake detection task. To this end, we rely on the FaceForensics++4 dataset (Rössler et al., 2019) consisting of 1,000 original video sequences (mostly frontal face without occlusions) that have been manipulated with four automated face manipulation methods.

Each video, either original or swapped is fed as input to the pyVHR pipeline; then, the estimated BVPs and the predicted BPMs can be analyzed in order to detect DeepFakes. It is reasonable to imagine that the BVP signals estimated on original videos would have much lower complexity if compared with the swapped ones, due to the stronger presence of PPG related information that would be possibly ruled out during swapping procedures. As a consequence, BVP signals from DeepFakes would perhaps exhibit higher levels of noise and hence more complex behaviour.

There exist many ways of measuring the complexity of a signal; here we choose to compute the Fractal Dimension (FD) of BVPs; in particular the Katz’s method (Katz, 1988) is employed.

The FD of the BVP estimated from the ith patch on the k-th time window (Dik) can be computed as Katz (1988): Dik=log10L/alog10d/a,

where L is the sum of distances between successive points, a is their average, and d is the maximum distance between the first point and any other point of the estimated BVP signal.

The FD associated to a given video can then be obtained via averaging: FD ˆvid=1PK∑i=0P ∑k=0KDik.

Similarly, one could consider adopting the average Median Absolute Deviation (MAD) of the BPM predictions on a video as a predictor of the presence of DeepFakes: MAD ˆvid=1K∑k=0KMADk.

Figure 18 shows how the FaceForensics++ videos lie in the 2-dimensional space defined by the average Fractal Dimension (FD ˆ) of predicted BVPs using the POS method and the average MADs of BPM predictions (MAD ˆ), when considering the original and swapped videos with the FaceShifter method.

Figure 17 Class diagram of dataset hierarchy of classes.

Figure 18 Deepfake detection results.

The 1,000 FaceForensics++ original videos (blue) and their swapped versions (yellow) represented in the 2-D space of BVP Fractal Dimension vs. BPMs average MAD. The green and red half-spaces are simply learned via a linear SVM.

It is easy to see how adopting these simple statistics on pyVHR’s predictions allows to discriminate original videos from DeepFakes. In particular, learning a baseline Linear SVM for the classification of Real vs. Fake videos generated by the FaceShifter method, yields an average 10-fold Cross-Validation Accuracy of 91.41% ± 2.05. This result is comparable with state of the art approaches usually adopting much more complex solutions.

Conclusions

In recent years, the rPPG-based pulse rate recovery has attracted much attention due to its promise to reduce invasiveness, while granting higher and higher precision in heart rate estimation. In particular, we have witnessed the proliferation of rPPG algorithms and models that accelerate the successful deployment in areas that traditionally exploited wearable sensors or ambulatory monitoring. These two trends, combined together, have fostered a new perspective in which advanced video-based computing techniques play a fundamental role in replacing the domain of physical sensing.

In this paper, in order to allow the rapid development and the assessment of new techniques, we presented an open and very general framework, namely pyVHR . It allows for a careful study of every step, and no less important, for a sound comparison of methods on multiple datasets.

pyVHR is a re-engineered version of the framework presented in Boccignone et al. (2020a) but exhibiting substantial novelties:

• Ease of installation and use.

• Two distinct pipelines for either traditional or DL-based methods.

• Holistic or patch processing for traditional approaches.

• Acceleration by GPU architectures.

• Ease of extension (adding new methods or new datasets).

The adoption of GPU support allows the whole process to be safely run in real-time for 30 fps HD videos and an average speedup (timeseq/timeparall) of around 5.

Besides addressing the challenges of remote Heart Rate monitoring, we also expect that this framework will be useful to researchers and practitioners from various disciplines when dealing with new problems and building new applications leveraging rPPG technology.

Additional Information and Declarations

Competing Interests

Author Contributions

Data Availability

1 Freely available on GitHub: https://github.com/phuselab/pyVHR.

2 Available at: https://sites.google.com/view/ybenezeth/ubfcrppg.

3 Available for download at https://github.com/partofthestars/LGI-PPGI-DB.

4 Available at GitHub: https://github.com/ondyari/FaceForensics.

The authors declare there are no competing interests.

Giuseppe Boccignone conceived and designed the experiments, performed the experiments, authored or reviewed drafts of the paper, and approved the final draft.

Donatello Conte and Giuliano Grossi analyzed the data, performed the computation work, authored or reviewed drafts of the paper, and approved the final draft.

Vittorio Cuculo performed the experiments, performed the computation work, prepared figures and/or tables, and approved the final draft.

Alessandro D’Amelio conceived and designed the experiments, performed the computation work, prepared figures and/or tables, and approved the final draft.

Raffaella Lanzarotti conceived and designed the experiments, analyzed the data, authored or reviewed drafts of the paper, and approved the final draft.

Edoardo Mortara performed the experiments, performed the computation work, prepared figures and/or tables, and approved the final draft.

The following information was supplied regarding data availability:

The code is available at GitHub: https://github.com/phuselab/pyVHR.

The UBFC data used in the experiments is available at: https://sites.google.com/view/ybenezeth/ubfcrppg.

The FaceForensics++ dataset used in the case study is available at GitHub: https://github.com/ondyari/FaceForensics.

The LGI-PPGI dataset is available at GitHub: https://github.com/partofthestars/LGI-PPGI-DB.

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
