# Peer review of "pyVHR: a Python framework for remote photoplethysmography"

_PeerJ Computer Science, doi:10.7717/peerj-cs.929_

## Round 0.1 · original submission · Major Revisions

Please consider the reviewers' comments in order to make the necessary major revisions.

Reviewer 1 ·

Basic reporting

Not all abbreviations are explained in text. For e.g. pyVHR, GPU, RGB, BVP ... Please check all abbreviations.

Experimental design

No comment.

Validity of the findings

No comment.

Reviewer 2 ·

Basic reporting

The topic for this study is not new but very interesting, and the methods are clearly described. The proposed system is important for health and biomedical applications. However, some major points are required before any progress.

Experimental design

# The technical approach isn't defined in detail (e.g., no mathematical equations and no justifications for design choices). The experimental design was prepared in good style.

Validity of the findings

Good

Additional comments

Article title: An end-to-end Python Framework for Remote Photoplethysmography
# Overall statement or summary of the article:
The topic for this study is not new but very interesting, and the methods are clearly described. The proposed system is important for health and biomedical applications. However, some major points are required before any progress.

# Please add some of the most important quantitative results to the Abstract and conclusion.
# In section 1.1, the authors should clearly mention the weakness point of former works (identification of the gaps) and shows the key differences between the different previous methods and the proposed method. A comparative overview table should solve this point.
# The technical approach isn't defined in detail (e.g., no mathematical equations and no justifications for design choices).
# Please describe your software algorithm in a flowchart or block diagram.
# The authors should describe more details for the used dataset in a table instead of mentioning the reference only.
# Please state whether the study was conducted in accordance with the Declaration of Helskinki (the author should provide the human ethics protocol number).

Reviewer 3 ·

Basic reporting

Main comments:

I commend the authors for this project. It would indeed be very valuable to have an open-source Python library available that allows for consistent comparison of state-of-the-art approaches on multiple datasets.

The main weakness of the project and this accompanying paper (in their current state) is that they seem to be stuck in 2018. Although the paper purports to be based on the most recent literature, there exists a whole slew of more modern, state-of-the-art rPPG methods (see detailed comments) and newer, larger datasets (e.g., VIPL-HR, OBF) which are unfortunately not addressed.

To be accepted for publication in its current form, the paper needs to at least acknowledge that it focuses on legacy rPPG methods (pre-deep learning period).

To make this project considerably more useful for the rPPG research community, the authors would future-proof the framework. This means relaxing the rPPG framework assumptions that existed until 2017 and supporting addition of modern rPPG models to the library, which take an entire stack of (cropped) video frames as input.

There are also several other minor issues in the detailed comments below.

Experimental design

Not applicable

Validity of the findings

Not applicable

Additional comments

Detailed comments:

l. 1: I got slightly confused about the usage of the phase “end-to-end” in the title and throughout this paper. It seems to be taken to mean “a method which goes from raw video all the way to to HR estimates” – but then shouldn’t all rPPG methods that estimate a HR from video be described as end-to-end? It could also be seen as also contentious because the phrase has a special meaning when it comes to deep learning.
l. 18: ... analyzing ...
l. 18: Sentence is too long. Break up into two sentences.
l. 20: Today, virtually all novel rPPG methods use machine learning to estimate the pulse signal or the heart rate. It is not obvious how those methods fit into the framework introduced in this paper.
l. 52: This framework assumes that the selection of the region of interest is external to the definition of rPPG methods; and that all rPPG methods are defined based on the averaged RGB traces. While this framework is applicable to the methods implemented here, it is not compatible with more modern learning-based rPPG methods that usually regard entire video frames (possibly after face detection) as input.
l. 70: real time
l. 118: The introduction for Section 3 confused me – two main building blocks are mentioned here, but the following subsection structure is not organized accordingly.
l. 123/Figure 2: Again, I believe the sequence of steps defined here do not apply “for the vast majority of rPPG methods proposed in the literature” – in fact, they probably only apply for very few methods proposed since 2018. Some examples of recent work that it does not apply to:
- DeepPhys: Video-Based Physiological Measurement Using Convolutional Attention Networks. https://openaccess.thecvf.com/content_ECCV_2018/html/Weixuan_Chen_DeepPhys_Video-Based_Physiological_ECCV_2018_paper.html
- RhythmNet: End-to-end Heart Rate Estimation from Face via Spatial-temporal Representation. https://arxiv.org/abs/1910.11515
- AutoHR: A Strong End-to-end Baseline for Remote Heart Rate Measurement with Neural Searching. https://arxiv.org/abs/2004.12292
- Multi-Task Temporal Shift Attention Networks for On-Device Contactless Vitals Measurement. https://arxiv.org/abs/2006.03790
- The Benefit of Distraction: Denoising Remote Vitals Measurements using Inverse Attention. https://arxiv.org/abs/2010.07770
- MetaPhys: few-shot adaptation for non-contact physiological measurement. https://arxiv.org/abs/2010.01773
- The Way to My Heart Is Through Contrastive Learning: Remote Photoplethysmography From Unlabelled Video. https://openaccess.thecvf.com/content/ICCV2021/html/Gideon_The_Way_to_My_Heart_Is_Through_Contrastive_Learning_Remote_ICCV_2021_paper.html
- PhysFormer: Facial Video-based Physiological Measurement with Temporal Difference Transformer. https://arxiv.org/abs/2111.12082
l. 241: See the previous point. The authors does not seem to be aware of new learning-based methods published since 2018.
l. 312: I like the detailed explanations here. Maybe the authors could use some of the information from here to qualify the statement about real-time applicability made earlier in line 70?
l. 322: In my own investigations, I found that after optimizing the algorithms, at some point the computation required for spatial averaging of the skin regions can become a bottleneck. Have any investigations been made as to how the performance differs when varying the resolution of the original raw video?
l. 355: Well done on including statistical assessments, these get ignored too often.
l. 356: outperform (without s)
l. 399: Many papers on rPPG also use the SNR (signal to noise ratio) metric. Is there any reason this wasn’t included here?
l. 580: In light of my previous comments, and to future-proof this framework, I think it would be useful to add support for rPPG methods which rely on an entire video (potentially previously cropped to include a face) as input with dimensions [T, H, W, C].
l. 668: As pointed out in earlier comments, framework is not general enough to accommodate recent work in the field.

---

## Round 0.2 · accepted · Accept

Thank you for your time and efforts in improving the quality of your submission according to the reviewer's comments.

Reviewer 3 ·

Basic reporting

The authors have addressed all my concerns. Thank you for this project and putting in the effort to future-proof it by supporting DL model.

Experimental design

The authors have addressed all my concerns. Thank you for this project and putting in the effort to future-proof it by supporting DL model.

Validity of the findings

The authors have addressed all my concerns. Thank you for this project and putting in the effort to future-proof it by supporting DL model.

Additional comments

The authors have addressed all my concerns. Thank you for this project and putting in the effort to future-proof it by supporting DL model.